# FGF23, Biomarker or Target?

**DOI:** 10.3390/toxins11030175

**Published:** 2019-03-22

**Authors:** Cristian Rodelo-Haad, Rafael Santamaria, Juan R. Muñoz-Castañeda, M. Victoria Pendón-Ruiz de Mier, Alejandro Martin-Malo, Mariano Rodriguez

**Affiliations:** 1Nephrology Service, University Hospital Reina Sofia, 14005 Cordoba, Spain; crisroha@yahoo.com (C.R.-H.); rsantamariao@gmail.com (R.S.); juanr.munoz.exts@juntadeandalucia.es (J.R.M.-C.); amartinma@senefro.org (A.M.-M.); marianorodriguezportillo@gmail.com (M.R.); 2Maimonides Institute for Biomedical Research of Cordoba (IMIBIC)/University of Cordoba, 14005 Cordoba, Spain; 3Spanish Renal Research Network (REDinREN), Institute of Health Carlos III, 28040 Madrid, Spain

**Keywords:** calcium, phosphate, chronic kidney disease, dialysis, fibroblast growth factor 23 (FGF23), fibroblast growth factor receptor (FGFR), *Klotho*, parathyroid hormone

## Abstract

Fibroblast growth factor 23 (FGF23) plays a key role in the complex network between the bones and other organs. Initially, it was thought that FGF23 exclusively regulated phosphate and vitamin D metabolism; however, recent research has demonstrated that an excess of FGF23 has other effects that may be detrimental in some cases. The understanding of the signaling pathways through which FGF23 acts in different organs is crucial to develop strategies aiming to prevent the negative effects associated with high FGF23 levels. FGF23 has been described to have effects on the heart, promoting left ventricular hypertrophy (LVH); the liver, leading to production of inflammatory cytokines; the bones, inhibiting mineralization; and the bone marrow, by reducing the production of erythropoietin (EPO). The identification of FGF23 receptors will play a remarkable role in future research since its selective blockade might reduce the adverse effects of FGF23. Patients with chronic kidney disease (CKD) have very high levels of FGF23 and may be the population suffering from the most adverse FGF23-related effects. The general population, as well as kidney transplant recipients, may also be affected by high FGF23. Whether the association between FGF23 and clinical events is causal or casual remains controversial. The hypothesis that FGF23 could be considered a therapeutic target is gaining relevance and may become a promising field of investigation in the future.

## 1. Introduction

Cardiovascular disease is the main cause of death in end-stage renal disease patients (ESRD) [1,2]. Mineral and bone disorders, vascular calcifications, and the vascular dysfunction associated with chronic kidney disease (CKD-MBD) are determinant in the outcomes of these patients [3,4,5]. CKD-MBD syndrome, which includes bone metabolic disease, alterations in mineral metabolism, and vascular calcifications, has been associated with poor quality of life, a higher rate of hospitalizations, and risk of fractures and death among patients with ESRD [6]. The decline in renal function produces alterations in serum phosphate, calcium, and PTH, and the development of vascular calcifications and left ventricle hypertrophy (LVH) [7,8,9,10]. It is estimated that about 10% of the worldwide population have some degree of renal dysfunction. The early recognition of kidney disease has become of high concern due to social and economic reasons [11,12,13]. Interventions to correct traditional risk factors of mortality have not reduced the mortality rate in ESRD patients, which continues to be very high. Therefore, it has been hypothesized that there are non-traditional risk factors that must be identified and targeted to improve patient outcomes [14,15,16].

The discovery of FGF23 has revolutionized our understanding of secondary hyperparathyroidism [17] and CKD-MBD in general. There is a close association between FGF23 and many unwanted outcomes such as a high incidence of infections, hospitalizations, myocardial infarction, heart failure, and death in CKD patients; therefore, it is a priority not only to understand the mechanisms whereby FGF23 signals are transduced but also the variables associated with the increase in FGF23 so therapeutic strategies can be activated [18,19,20,21]. 

Overall, FGF23 plays a key role in the complex interrelationship between the bones and other organs. Today, the bones are recognized as an endocrine organ because they secrete hormones that allow crosstalk with other organs [22]. FGF23 is a hormone produced and secreted by different tissues, although the main source are the osteocytes and mature osteoblasts [23,24,25]. FGF23 increases urinary excretion of phosphate and reduces renal production of 1,25 dihydroxy-vitamin D (1,25(OH)_2_D_3_) [26,27,28]. FGF23 also decreases PTH secretion [29,30]. However, in recent years, FGF23 excess has been shown to produce detrimental effects in other organs such as the heart, bone structure, and endothelium (Figure 1) [3,31,32]. The fact that FGF23 may cause undesirable effects raises the question of whether it may not only be a biomarker of altered mineral metabolism and cardiovascular disease, but a therapeutic target [30]. This review summarizes our current understanding of FGF23, its regulation, and recognized effects on different organs. 

## 2. Secondary Hyperparathyroidism

Comprehensive reviews of SHP have been previously published [7,33,34] and are outside of the scope of this review. Briefly, in the old trade-off hypothesis, PTH and vitamin D were considered the hallmarks of secondary hyperparathyroidism. The progressive decrease in renal function resulted in phosphate accumulation and reduction of calcitriol production by the failing kidneys. Phosphate retention, low calcitriol levels, and the tendency to hypocalcemia stimulated PTH secretion, which promoted calcitriol production and inhibited calcium excretion through the kidneys, and also increased calcium and phosphate efflux from the bones [33,35,36]. Although this mechanism helps to maintain serum calcium levels, it is not sufficient to prevent phosphate accumulation. Serum phosphate was shown to stimulate PTH secretion and parathyroid cell proliferation in both animal models and humans [37,38]. The accumulation of phosphate contributes to hypocalcemia by producing skeletal resistance to the calcemic action of PTH [39]. The discovery of the FGF23/*Klotho* complex [40,41] helped to clarify the pathophysiology of secondary hyperparathyroidism. FGF23, together with the PTH‒vitamin D axis, configures one of the most advanced endocrine networks that manage communication between the bone and other organs [7]. In the early stages of CKD, FGF23 increases to maintain serum phosphate within normal levels even when PTH is still normal [42]. Once CKD progresses, these compensatory mechanisms fail, and secondary hyperparathyroidism becomes evident. Finally, in advanced CKD, hyperphosphatemia and hypocalcemia is present because the marked reduction of glomerular filtration makes FGF23 and PTH non-operative [29,43].

## 3. FGF23 Origin and Structure

The fibroblast growth factors (FGFs) belong to a family of proteins involved in embryonic development and metabolic functions [44,45]. All of them derive from the common ancestral gene, *Fgf13-like*, and are comprised of 22 structurally and evolutionarily similar members from *Ffg1* to *Fgf23* that conserve a ~120-residue structural domain [46]. Remarkably, *Fgf15* and *Fgf19* are ortholog proteins in vertebrates, so they are absent in humans and mice, respectively [44,47]. 

Phylogenetically, the FGFs family may be divided into seven different gene subfamilies that are grouped into three different subgroups according to their functions: the intracrine, the paracrine or canonical and the endocrine *Fgf* genes. The intracrine group includes proteins *Fgf11* to *Fgf14*, which act intracellularly through a pathway independent of the FGF receptor (FGFR) [45]. The canonical subgroup acts in an autocrine and paracrine manner through the binding and activation of the tyrosine kinase FGFR, which includes heparin/heparan sulfate as a cofactor. This group includes five subfamilies: *Fgf 1/2/5*, *Fgf 3/4/6*, *Fgf 7/10/22*, *Fgf 8/17/18*, and *Fgf 9/17/20* [46]. Finally, the endocrine group, comprised of *Fgf 19/21/23*, acts systemically in a hormonal manner through both an FGFR-dependent and FGFR-independent pathway [44]. In contrast to the canonical group, the endocrine family uses a COOH-terminal domain to activate FGFR, whereby they are not captured by the extracellular matrix so they can act as circulating factors [44]. 

The *Fgf23* gene is located on human chromosome 12p3.3, and is comprised of three separate exons and two introns that codify a 32 kDa glycoprotein with 251 amino acids. This full-length protein is recognized as a biologically active hormone, although some studies have suggested that c-terminal fragments may also have biological activity [48,49]. The COOH-terminal domain (c-terminal; 12 kDa) acts as a cofactor by inhibiting iFGF23 binding to the *FGFR/Klotho* complex [44,50]. Once the mature protein is released into the circulation, it can be measured as two different isoforms, iFGF23 (^25−^FGF23^−251^) and the c-terminal FGF23 (^25−^FGF23^−179^) [46]. Commercially available assays quantify circulating FGF23 levels based on the different epitopes expressed. Assays detecting iFGF23 recognize two epitopes beyond the proteolytic site. By contrast, assays detecting cFGF23 fragments recognize both iFGF23 and cFGF23 fragments because of the two epitopes captured distal to the cleavage site [50]. The simultaneous determination of both molecules allows for assessing the production and cleavage of the molecule [50]. 

## 4. Mechanisms of Action of FGF23

The main functions of FGF23 are to decrease the serum levels of 1,25(OH)_2_D_3_ through the inhibition of 1α-hydroxylase and increase 24-hydroxylase activity [51]. Moreover, it enhances phosphaturia by inhibiting phosphate proximal tubular resorption through sodium phosphate cotransporters NaPi2a and NaPi2b [52]. Similarly, PTH also regulates renal urinary phosphate excretion by promoting the internalization of NaPi2 cotransporters from the brush border membrane in renal proximal tubules [53]. In early stages of CKD, the increase in PTH is caused in part by a deficiency of 1,25(OH)_2_D_3_. The elevation of FGF23 reduces 1,25(OH)_2_D_3_ levels by decreasing renal production and also increasing catabolism. This may explain why in early CKD the elevation in serum PTH is observed once FGF23 is already increased [53]. FGF23 tissue-specific functions are dependent on the presence of FGF receptor (FGFR) and in some cases its cofactor α*Klotho* [41]. Four different FGFRs have been recognized, FGFR 1 to 4. Based on the distribution of these different receptors, FGF23 targets the kidneys, the parathyroid gland, the liver, the heart, the bone, the immune system, and possibly others [45,46]. *Klotho* gene encodes a 1014 amino acids type I transmembrane protein with β-glucuronidase activity composed of two extracellular domains, termed KL1 and KL2 [41], and is predominantly expressed on the kidney and the choroid plexus, although it has also been described in the parathyroid gland, the pituitary gland, placenta, skeletal muscle, pancreas, and testis, among others [54,55,56]. *αKlotho* was considered mandatory for FGF23 signaling recognition; however, the discovery of *Klotho*-independent pathways for FGF23 transduction has changed after the identification of FGFRs that do not always require *αKlotho* as a cofactor [57]. *αKlotho* and FGFRs are also abundantly expressed in the parathyroid glands [58]. Under physiological conditions FGF23 inhibits PTH secretion and production [58,59]; however, in uremia, FGF23 fails to inhibit PTH release because of downregulation of the parathyroid *FGFR1/Klotho* complex [60]. Hence, in ESRD patients on dialysis, FGF23 levels predict refractory SHP [61].

The molecule *αKlotho* is essential not only because its loss of function is associated with premature aging and target-organ resistance to FGF23 actions, but also because, in ESRD, *Klotho* depletion is considered an early biomarker of disease progression, development of vascular calcifications, and LVH [62,63]. The full-length *αKlotho* extracellular domain can be cleaved and secreted into the circulation as a soluble isoform, namely soluble *Klotho* (*sKlotho*), which not only enhances FGFR binding capability to FGF23 by 20-fold [64] but also acts as a scaffold to allow closer proximity between FGFR and FGF23, increasing the stability of the complex [65]. Some authors have suggested that the serum and urine levels of *sKlotho* may serve as a surrogate marker of renal *αKlotho* expression [66]. Although strong evidence is still lacking, *sKlotho* could be considered an endocrine mediator targeting different organs without *αKlotho* expression [57]. Interestingly, in the mouse, *sKlotho* protects the heart from uremic cardiomyopathy [67] and stress-induced hypertrophy and remodeling; this is achieved by *sKlotho*-induced inhibition of the transient receptor potential cation channel, subfamily C, member 6 (TRPC6) [68]. A number of reports show that *Klotho* deficiency correlates with the development of coronary artery disease, atherosclerosis, myocardial infarction, and left ventricular hypertrophy [31,69]. Therefore, *Klotho* may be involved in the regulation of signaling pathways and cell metabolism, being a key factor in cardiac and vascular protection. Furthermore, *sKlotho* is likely to be protective in rodent models of acute kidney injury (AKI), reducing renal fibrosis and CKD progression [66].

## 5. Regulation of FGF23 Production

Since the discovery of FGF23, many studies have been conducted aiming to evaluate the factors regulating FGF23 production and cleavage. 

### 5.1. Vitamin D

The administration of vitamin D increases FGF23 in both humans and rodents [70,71]. Furthermore, 1,25(OH)_2_D_3_ (calcitriol) increases intestinal absorption of phosphate and calcium, both of which also favors FGF23 production. Vitamin D by acting on its specific receptor (VDR) stimulates the promoter region in the FGF23 gene, an effect that is independent of serum phosphate and calcium [72]. Additionally, locally produced vitamin D in bone cells is likely to regulate FGF23 production [72]. 

### 5.2. Phosphate

Phosphate sensing receptors are yet to be discovered [73], but it is evident that phosphate load, even without high serum phosphate, stimulates FGF23 production [74,75]. In CKD patients and patients on dialysis, the high serum phosphate concentration is associated with elevated FGF23 levels [21,76]. We recently demonstrated that phosphate contributed 70% to the high levels of intact FGF23 in hemodialysis patients [21]. 

### 5.3. Calcium and PTH

PTH promotes the transcription of FGF23 in a calcitriol-independent manner [59,77]. Recently, the demonstration that PTH activates the orphan nuclear receptor Nurr1 [78] has revealed the putative mechanism by which PTH increases FGF23. In dialysis patients, FGF23 correlates positively with serum levels of phosphate and PTH, and inversely with the serum calcium concentration [21,79].

It is likely that hypocalcemia reduces circulating FGF23 to prevent a decrease in calcitriol, which would worsen a situation of calcium deficiency [80]. VDR null mice fed a high-calcium diet had increased circulating FGF23, which suggests that the regulation of FGF3 by calcium is independent of VDR [81]. In the general population and CKD patients, consumption of an enriched calcium diet is associated with increased FGF23 [82]. In dialysis patients, the calcium effect on FGF23 production is more prominent if phosphate is within the normal range [21]. A decrease in calcium below 8 mg/dL diminishes the elevation of FGF23 induced by high phosphate [80,83]. The reduction of PTH after the administration of calcimimetics is associated with a decrease in FGF23, which is likely related to the concomitant reduction in both serum calcium and phosphate [84]. 

### 5.4. Inflammation and Iron Deficiency

There may be bi-directional crosstalk between inflammation and FGF23 production [85]. Since both inflammation and elevation of FGF23 are associated with mortality, the understanding of this loop is of high relevance in clinical practice [86,87,88]. Either acute or chronic inflammation may promote FGF23 transcription and cleavage [89]. The induction of acute inflammation in wild-type mice not only reduced the serum iron and increased serum ferritin levels but also resulted in an increase in bone *Fgf23* mRNA expression that was accompanied by a rise in cFGF23, while iFGF23 remained unchanged [89]. By contrast, the induction of chronic inflammation in a murine CKD model was followed by a concurrent increase in both iFGF23 and cFGF23, although the rise in c-terminal fragments was greater than that of iFGF23 [89]. Mechanistically, inflammation and iron deficiency increase the activity of the hypoxia-inducible factor 1-alpha (HIF-1α) signaling, thereby augmenting *Fgf*23 transcription [89]. Therefore, inflammation and iron deficiency promote not only FGF23 transcription but also cleavage. In healthy women with iron deficiency, the administration of intravenous iron reduced cFGF23, whereas iFGF23 increases transiently, probably due to the reduction of FGF23 cleavage [90].

In CKD and dialysis patients, the high serum phosphate correlates with increased levels of FGF23, which in turn is associated with an elevation of C-reactive protein [21,91]. Inflammation may not only modulate FGF23 production but also cleavage since the relative effect of inflammation on the elevation of cFGF23 is 3-fold higher than that of iFGF23 [21]. A mechanism whereby inflammation may increase FGF23 production is the activation of the nuclear factor kappa-light-chain-enhancer of B-cells (NF-κB) pathway, which has been related to FGF23 transcription [24]. However, NF-κB may also enhance FGF23 production through the upregulation of HIF-1α, which is known to increase FGF23.

### 5.5. Erythropoietin

A few studies have demonstrated an association between FGF23 and erythropoiesis [92,93,94]. Although elevated FGF23 reduces bone marrow EPO expression, EPO promotes FGF23 transcription [93,94]; the precise mechanism remains uncertain, although it is likely to be independent of iron and *Klotho*. The administration of EPO to patients with AKI was followed by an increase in circulating FGF23 [94]. These results are supported by other additional studies in kidney transplant recipients and dialysis patients [95].

### 5.6. Others

#### 5.6.1. Adiponectin

The adipose tissue secretes many factors with endocrine functions. Indeed, adiponectin limits renal damage and accelerates renal recovery after kidney injury in mice models [96]. Chronic kidney disease upregulates adiponectin expression [97]. Also, adiponectin increases the renal excretion of calcium and therefore the risk of osteoporosis [98]. Recently, adiponectin has been coupled with systemic mineral homeostasis and renal handling of phosphate and calcium. Mechanistically, high adiponectin reduces kidney secretion of *αKlotho* and FGF23 production by osteocytes, although it increases the renal loss of calcium [99]. This mechanism is mediated through the activation of renal ADIPOR1 and ADIPOR2 receptors [99]. 

#### 5.6.2. Insulin

Diabetes is associated with elevated FGF23 [100]. Furthermore, increased cFGF23 has been associated with insulin resistance, obesity, resistin, and HOMA-IR [101]. However, these associations may be mediated through inflammation since these patients also showed higher levels of IL-6, CRP, and IL-10 [101]. Interestingly, a recent study has delineated insulin-dependent signaling for FGF23 synthesis. Insulin and insulin-*like* growth factor (IGF-1) suppress FGF23 production by activating the PI3K/PKB/Akt/FOXO 1 signaling [102].

#### 5.6.3. Aldosterone

One remarkable finding in recent years has been the discovery of the crosstalk between FGF23 and aldosterone regulation. By activating the FGFR1/α-*Klotho*/ERK pathway, FGF23 enhances sodium reabsorption through sodium/chloride cotransporter in distal tubules with a significant impact on volume overload and an elevation of blood pressure [103]. In parallel, aldosterone upregulates FGF23 secretion, which in turn increases the expression of angiotensin II in cardiac myocytes [104]. In dialysis patients, volume overload is associated with elevated FGF23 [105]. 

#### 5.6.4. Regulation of FGF23 Production by Bone Cell Factors

Production of FGF23 is regulated at the bone level by the gen with homology to endopeptidases located in the X chromosome (PHEX) and the dentin matrix protein 1 (DMP1). PHEX is a 106 kDa protein expressed by osteoblast and osteocytes. Inactivating the mutation of PHEX results in excessive *Fgf23* gene transcription. On the contrary, overactivation of PHEX decreases FGF23. It seems that the cleavage of the intact protein is the main regulating function of PHEX [58,106]. The DMP1 is a 94 kDa protein expressed in osteoblasts and osteocytes that is critical in bone mineralization [24]. Even though its inhibition resembles PHEX inactivation, its overactivation does not contribute to FGF23 elevation [46]. 

## 6. Effect on Different Organs

Classically, the kidneys and the parathyroid glands are the main targets for FGF23. However, by acting in a *Klotho*-independent manner, high FGF23 has been demonstrated to have other effects that in some cases are detrimental (Figure 1). Hence, one question to be addressed is whether FGF23 should be a therapeutic target. The effects of FGF23 on the kidneys and parathyroid are well described; the present manuscript will focus on the often detrimental effects of FGF23 in other organs. Table 1 summarizes the classical and non-classical effects of FGF23, the different targeted cell types, and the FGFR isoform and final organ effect induced.

### 6.1. The Heart

Left ventricular hypertrophy (LVH) is frequently observed in patients with advanced CKD [107]. More recently, FGF23 has also been associated with a deleterious effect on the heart [108] through a *Klotho*-independent manner since the heart lacks α*Klotho* expression [31]. Hence, FGF23 targets the heart directly by acting on FGFR4 with subsequent triggering of the PLCγ/calcineurin/NFAT signaling pathway, which promotes cardiac hypertrophy, fibrosis, and heart failure [31]. The latter effect has been shown in in vitro studies and non-CKD mice models [31]. Inhibitors for FGFR4 halt such a detrimental effect [109]. As described above, the FGF23 increase is accompanied by reductions in *Klotho*. Reductions in *sKlotho* per se are also associated with cardiac injury in animal models [67]. Indeed, the use of recombinant *sKlotho* in a CKD model of *Klotho*-deficient mice attenuated cardiac remodeling regardless of the prevailing FGF23 levels [67,110]. Hence, there is evidence suggesting that LVH is due to a *Klotho* deficiency rather than to an FGF23 excess since the normalization of both phosphate and FGF23 did not prevent the development of LHV in *Klotho*-replete cultures of cardiac myocytes [67,110]. Studies inducing isolated *sKlotho* reductions in the absence of an FGF23 effect by blocking FGFR4 may serve as models to demonstrate whether *sKlotho* actions of cardiac injury development are dependent on FGF23 increase or not. 

It is also important to consider that FGF23 may also increase blood pressure [103], inflammation [89], and CKD progression [111], all of which are associated with the development of LVH. However, it is likely that FGF23 is such a potent cardiac remodeling molecule that its effect on cardiac cells is independent of other pro-hypertrophic factors [57]. Interestingly, the cardiac expression of FGF23 and FGFR4 is elevated in human cardiomyocytes from deceased dialysis patients [112]. In these patients, cardiac FGF23 correlate with cardiac myocytes’ cross-sectional area and with brain natriuretic peptide (BNP). Although *αKlotho* is absent in cardiac myocytes, *sKlotho* has been detected in lysates from human myocardial tissue [112]. However, *sKlotho* was lower in patients with LVH and its levels correlated negatively with ESRD duration and the type of renal replacement therapy. Interestingly, recent observations have shown that the length of the exposure to an increment in FGF23 is more critical for cardiac remodeling than the magnitude of FGF23 elevation [108,113].

Regarding FGF23-associated myocardial fibrosis, early studies showed that FGF23 promotes the expression of TGF-β in cardiac fibroblasts, which induce cardiac fibrosis through the β-catenin pathway [114]. FGF23 also stimulates the proliferation of mice cardiac fibroblasts and collagen I and II synthesis, at least after myocardial infarction [115]. In myocardial tissue from dialysis patients, cardiac fibrosis correlates positively with the dialysis vintage and negatively with the highly prevalent deficiency of *sKlotho* in cardiac cells [116]. Such an effect may be prevented by the injection of recombinant *sKlotho* [67] as it inhibits the transient receptor potential cation channel, subfamily C, member 6 (TRPC6) in the mouse heart [68]. Hence, the existence of multiple pathways by which FGF23 acts over the cardiac cells warrants further research to evaluate whether neutralizing the FGFR4 has any beneficial effect on LVH [57].

Finally, according to some reports, all effects of FGF23 on the heart may not be that harmful; FGF23 may increase intracellular calcium in cardiac myocytes, promoting contractility, although this may contribute to a higher risk of arrhythmia [117,118].

### 6.2. Liver

The liver is one of the organs with the highest expression of FGFR4 [55,119]. Similarly to the heart, FGF23 targets the liver through the activation of the FGFR4/PLCγ/calcineurin/NFAT pathway [85]. Following the activation of this signaling pathway, FGF23 stimulates the production of inflammatory cytokines such as C-reactive protein (CRP), IL-6, IL-12, and TNFα [85]. The Kupffer cells, which produce inflammatory cytokines, respond to FGF23 stimulation [120]. Interestingly, FGF23 is likely to promote hepatocyte proliferation [57] and does not seem to produce hepatic tissue injury since there is no increase in liver enzymes [57]. However, a recent study reported that FGF23 and reduced circulating calcifediol were independently associated with non-alcoholic fatty liver disease, perhaps because of its association with insulin resistance and diabetes [121]. 

Patients with end-stage renal disease (ESRD) have inflammation due to several reasons, including the dialysis procedure [122]. High levels of FGF23 stimulate cytokine production, which mediates systemic inflammation [85]. Altogether, FGF23, ESRD, and the cytokines being secreted are independently associated with cardiovascular disease. As in the heart, FGFR4 could be pharmacologically targeted by the use of FGFR4 inhibitors [109]. Further studies are required to define appropriate strategies to reduce FGF23-mediated cytokine production.

### 6.3. Immune System

Infections are more frequent in patients with kidney failure than in the general population [130]. Recently, FGF23 has been associated with a higher proportion of infections in ESRD patients [18,125]. FGF23 inhibits the activation, binding, and migration of neutrophils by preventing β2-integrin activation [125]. This effect is likely to be *Klotho*-independent and is mediated by FGFR2, although FGFR1 and FGFR4 are also expressed in neutrophils [131]. Also, it has been shown that FGF23 reduces the expression of CD11b integrin, thus reducing neutrophils chemotaxis [109,132]. Importantly, the effect of recombinant FGF23 on neutrophils in rodent models is dose-dependent and is mainly observed with high concentrations of FGF23, as high as those observed in dialysis patients [18,125]. Furthermore, FGF23 also upregulates TNFα production and downregulates calcitriol production by monocytes [120,133]. 

### 6.4. Skeleton

FGF23 may affect bone mineralization in both a *Klotho*-dependent and independent manner. In a *Klotho*-independent manner, FGF23 inhibits tissue-nonspecific alkaline phosphatase (TNAP) transcription in osteoblasts, which is crucial in bone mineralization [32]. However, other studies have shown that in an FGFR1/*Klotho*-complex-dependent manner, FGF23 suppresses TNAP in mouse osteoblast-like cells [127]. Further studies should clarify the factors that make one mechanism overcome the other. Interestingly, *Klotho* deletion in osteocytes increases bone formation and bone mass [128]. 

### 6.5. Bone Marrow and Anemia

Although CKD-related anemia may be multifactorial, FGF23 has been associated with reduced erythrocyte production and differentiation [92]. FGF23 decreases erythropoiesis and erythropoietin (EPO) production [92,129]. This effect can be neutralized by blocking FGF23 signaling with FGF23 blocking peptides [92]. In CKD patients, circulating levels of FGF23 are associated with a decrease in hemoglobin levels [134]. The previously described effect of FGF23 on erythropoiesis may be *Klotho*-independent since there is no evidence of *αKlotho* expression in bone marrow cells [55].

### 6.6. Other Organs

Other effects of FGF23 have been described in organs such as the brain, lungs, skeletal muscle, and endothelium, but the mechanisms involved have not been elucidated.

Different types of brain cells possess complex FGFR1-*αKlotho*, but their function remains uncertain [57]. An increase in FGF23 levels is associated with reduced neuronal ramifications and enhanced synaptic density; this is mediated by the activation of PLCγ signaling and takes place in the absence of *αKlotho*. This effect may be transduced in memory deficits [135]. However, it seems that a high phosphate intake may reverse this effect, challenging the hypothesis of a potentially detrimental role of FGF23 on memory cells [136].

Patients on dialysis often complain of muscle weakness. Indeed, sarcolemma shortening is observed in ESRD patients [57,137]. In animal models, FGF23 is associated with a reduction of muscle strength [138], an effect that should be mediated by FGFR4 since, like the heart muscle, skeletal muscle lacks the expression of *Klotho*. Reports on the effects of FGF23 on skeletal muscle are not uniform since other studies indicate that high FGF23 may favor exercise performance [139].

Although the lung tissue expresses FGFR1, FGFR2, FGFR3, and FGFR4, to our knowledge there is no strong evidence indicating that FGF23 have some specific effects on the lungs. It is likely that FGF23 increases IL-8 secretion by epithelial cells, although there is no evidence of a harmful effect [140]. Some evidence points to a lack of *Klotho* as a risk factor for the development of lung emphysema [141], but the evidence is scarce and needs further investigation. 

*sKlotho* expression protects the endothelium from uremic-induced aging [142]. However, results obtained from studies evaluating the effect of FGF23/*Klotho* on the endothelium are conflicting. Whereas some studies have associated FGF23 with endothelial dysfunction [143], the data in CKD patients are controversial [144,145]. Perhaps studies aiming to analyze the potential effect of FGF23 on vascular smooth muscle cells would help to define the direct effect of FGF23 on the vascular wall.

## 7. Clinical Impact of FGF23

Chronic kidney disease is the main cause of a secondary increase in FGF23 level [27]. Starting in the early stages of CKD, an increase in FGF23 enhances the urinary excretion of phosphate, which compensates for the reduced filtration of phosphate [42]. Of note, the kidneys metabolize FGF23 since the values obtained in the renal vein are less than in the renal artery [25]. Together with phosphate, FGF23 is independently associated with the progression of kidney disease [111,123]. Hence, FGF23 is not only the consequence of CKD but also a cause of CKD progression. In dialysis patients, FGF23 levels may reach extremely high levels. Following kidney transplantation, FGF23 decreases rapidly, together with a recovery of *Klotho* [146]. Nonetheless, FGF23 is associated with a high mortality rate in this population; however, there are many residual confounders, not easy to dissect out, that are also associated with adverse outcomes [147].

Regardless of the population evaluated, high FGF23 levels are associated with adverse clinical effects such as cardiovascular disease and mortality [148,149]. The strongest evidence comes from patients in dialysis, in whom, independent of the confounders included in the Cox proportional analysis, FGF23 remains associated with a high mortality rate [84,88,150]. These results are not surprising given the association of FGF23 with the development of LVH [31], coronary artery disease and myocardial infarction [151], stroke [152], impairment of the immune response [125] and infection-associated death [18]. Thus, FGF23 should be considered as a relevant uremic toxin [153]. Longitudinal analysis of FGF23 over time has shown that those patients with upward trends in circulating FGF23 have the highest risk of poor outcomes [154]. The harmful effects of high FGF23 levels on different organs such as the heart, bones, liver, and immune system lead us to question whether FGF23 is a potential therapeutic target.

A recently published study suggests that FGF23 elevation in the absence of CKD is not causative of cardiovascular disease [145]. Indeed, some authors suggest that FGF23 may be a consequence rather a cause of cardiovascular disease [155], since patients from the general population may show a comparable risk of death to that of dialysis patients [20], suggesting casualty rather than causality. In this line, demonstrating that a decrease in adverse outcomes follows ensures the reduction in FGF23 would be crucial.

## 8. Targeting FGF23

Given the broad list of effects associated with the high levels of FGF23, it would be reasonable to think of FGF23 as a clinical target. If FGF23 is increased, a reduction in phosphate would be the first step to decrease FGF23 [156]. In dialysis patients, dietary interventions should be the first-line treatment for hyperphosphatemia; however, very restricting diets in this population may translate into malnutrition. Thus, phosphate binders are necessary for phosphate control. The use of calcium-free phosphate binders helps to reduce FGF23 levels [21,157,158], whereas calcium-containing binders are likely to increase FGF23, another reason to limit the use of calcium-containing compounds [159]. Interestingly, post hoc analysis of the EVOLVE study demonstrates that cinacalcet reduces not only PTH but also FGF23 and cardiovascular events [84]. 

Burosumab, a new monoclonal antibody that targets FGF23, has shown benefits in patients with X-linked hypophosphatemia [160]. It remains unclear whether the complete blocking of FGF23 would be detrimental in CKD not on dialysis. Experimental studies have shown that the neutralization of FGF23 causes elevation of phosphate and vitamin D, resulting in vascular calcification [161]. Presumably, in dialysis patients, neutralization of FGF23 should not be detrimental. The option of selective blocking of FGFR receptor deserves investigation.

To date, there is no evidence of improved outcomes in patients following FGF23 reduction. Longitudinal studies evaluating the change in FGF23 levels are warranted to evaluate whether the reduction in FGF23 is associated with a decrease in those adverse effects being attributed to FGF23.

In summary, much has been learned about the regulation and effects of FGF23. Presently, the adverse effects attributed to FGF23 are many. It is important to learn how many of these effects are so undesirable as to warrant a strategy directed at actively reducing the actions of FGF23 by either neutralization of the molecule or blockade of the receptors. 

## Figures and Tables

**Figure 1 toxins-11-00175-f001:**
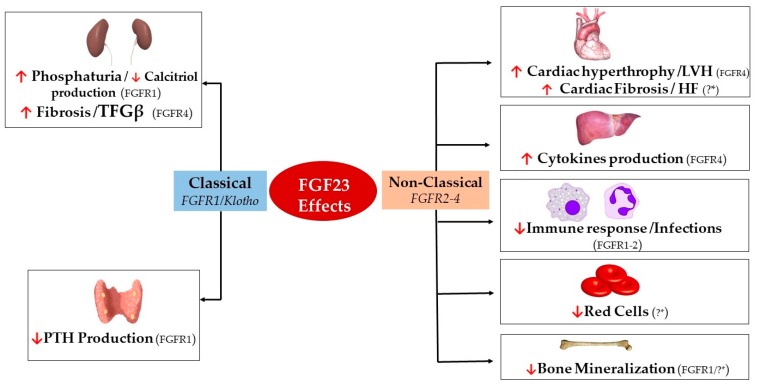
Final classical and non-classical clinical effects of FGF23 on different organs. TFGβ, transforming growth factor βeta; PTH, parathyroid hormone; LVF, left ventricle hypertrophy; HF, heart failure. ↓ Decrease; ↑ Increase; ?* Presumably *Klotho-*independent.

**Table 1 toxins-11-00175-t001:** Non-classical effects of FGF23 on different organs.

	Effect	Organ Target	Cell Type	FGFR Isoform	Organ Effect
FGF23	Classical *(FGFR1/Klotho)*	Kidneys [25,123]	Renal Tubular Epithelial Cells	FGFR1	↑ Phosphate Excretion ↓ Calcitriol production
			Renal Fibroblasts	FGFR4	↑ TFGβ/Fibrosis
		Parathyroid Glands [58,59,77]	Parathyroid Chief Cells	FGFR1	↓ PTH Excretion
	Non-classical *(FGFR2-4)*	Heart [31,109,112,114,124]	Cardiac Myocytes	FGFR4	Hypertrophy/LVH
			Cardiac Fibroblasts	?*	Cardiac Fibrosis/HF
		Liver [85]	Hepatocytes	FGFR4	↑ IL-6/CRP Secretion
		Immune System [120,125,126]	Neutrophils	FGFR2	↓ β-2 Integrin Activation
			Macrophages	FGFR1	↑ TNFα Production
		Skeleton [32,127,128]	Osteocytes/Osteoblasts	FGFR1/?*	↓ TNAP Transcription
		Bone Marrow [92,129]	Early Erythroid Progenitors/BFU-E Colonies	?*	↓ Red Cells

FGF23 targets different cell types across different organs. Although the main mechanistic effects are *Klotho*-dependent, off-target effects are in some cases *Klotho*-independent based on the involvement of FGFR isoforms leading to tissue-specific effects. ↓ Decrease; ↑ Increase; ?* Presumably *Klotho*-independent. BFU-E colonies, colony forming for erythroid progenitors.

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
