# Peer review of "FGF23, Biomarker or Target?"

_toxins, 2019, doi:10.3390/toxins11030175_

Round 1
Reviewer 1 Report
This review was well summarized about FGF23 of current understanding, its regulation, effects on different organs (heart, liver, immune system, skeleton, bone marrow and anemia, other organs), briefly and precisely, with recent good references. And authors were also discussed the clinical impact of FGF23 and targeting for FGF23. I believe that this review gives readers learning knowledge and gives hints for further research. So, I evaluated that this review will be worth for publication in this journal.
Author Response
We are deeply thankful with the reviewer for his work that has made our review improve.
Reviewer 2 Report
In this review author describe about the bone derived FGF-23, that plays as a major regulator of phosphate homeostasis which might appears to be an important biomarker for patients with chronic kidney disease (CKD), positively associated with obesity associated type 2 diabetes (T2D) and cardiovascular disease (CVD). Therefore, targeting FGF-23 could be a therapeutic for CKD. Overall this review is well written. Importantly, this review provides critical insight into the role of FGF-23 in progression of CKD and end stage renal disease. However, there are few issues that should be addressed as outlined below.
1) Mechanism of action of PTH and FGF-23 are similar in renal tubules in terms of phosphate homoeostasis- they reduce the expression of sodium dependent phosphate co-transporters (NPT2a and NPT2c) but in very early stage of CKD circulating levels FGF-23 increases whereas level of PTH is in normal range despite of same mechanism. Author should explain this discrepancy.
2) Author should discuss the positive association of circulating FGF-23 with non-alcoholic fatty liver disease (NFLAD) which is independent of serum 25-hydroxyvitamin D
(Xingxing He, Yun Shen, Xiaojing Ma, Lingwen Ying, Jiahui Peng, Xiaoping Pan, Yuquian Bao, Jian Zhou.The association of serum FGF23 and non‐alcoholic fatty liver disease is independent of vitamin D in type 2 diabetes patients. Clin.Exp Pharmacol Physiol and 2018 Jul: PMID:29574933.)
3) Author should improve introduction part.
4) Author should write abbreviate of NaPi2a, and NaPi2b
Author Response
1) Mechanism of action of PTH and FGF-23 are similar in renal tubules in terms of phosphate homoeostasis- they reduce the expression of sodium dependent phosphate co-transporters (NPT2a and NPT2c) but in very early stage of CKD circulating levels FGF-23 increases whereas level of PTH is in normal range despite of same mechanism. Author should explain this discrepancy.
- We agree with the reviewer in such an important issue. In consequence, we have added information in line 115 to 120.
2) Author should discuss the positive association of circulating FGF-23 with non-alcoholic fatty liver disease (NFLAD) which is independent of serum 25-hydroxyvitamin D
(Xingxing He, Yun Shen, Xiaojing Ma, Lingwen Ying, Jiahui Peng, Xiaoping Pan, Yuquian Bao, Jian Zhou. The association of serum FGF23 and non‐alcoholic fatty liver disease is independent of vitamin D in type 2 diabetes patients. Clin.Exp Pharmacol Physiol and 2018 Jul: PMID:29574933.)
- Information regarding the potential effect of FGF23 on non-alcoholic fatty live has been added on the main text and referenced on accordance. Line 312 to 314.
3) The author should improve the introduction part.
-Information included in the introduction section has been improved.
4) The author should write abbreviate of NaPi2a and NaPi2b
- Abbreviate for Napi2a, and Napi2b can be read in line 114 to 115.